# Research Methods and New Advances in Drug–Drug Interactions Mediated by Renal Transporters

**DOI:** 10.3390/molecules28135252

**Published:** 2023-07-06

**Authors:** Kexin Lin, Xiaorui Kong, Xufeng Tao, Xiaohan Zhai, Linlin Lv, Deshi Dong, Shilei Yang, Yanna Zhu

**Affiliations:** Department of Pharmacy, First Affiliated Hospital of Dalian Medical University, Dalian 116011, China; linkexin2022@163.com (K.L.); kxrxiaorui@163.com (X.K.); taoxufeng.2008@163.com (X.T.); hanhanjiayoudl@163.com (X.Z.); lvlinlinyu@163.com (L.L.); deshidong@163.com (D.D.)

**Keywords:** renal transporter, drug–drug interactions, analytical tools, model

## Abstract

The kidney is critical in the human body’s excretion of drugs and their metabolites. Renal transporters participate in actively secreting substances from the proximal tubular cells and reabsorbing them in the distal renal tubules. They can affect the clearance rates (CLr) of drugs and their metabolites, eventually influence the clinical efficiency and side effects of drugs, and may produce drug–drug interactions (DDIs) of clinical significance. Renal transporters and renal transporter-mediated DDIs have also been studied by many researchers. In this article, the main types of in vitro research models used for the study of renal transporter-mediated DDIs are membrane-based assays, cell-based assays, and the renal slice uptake model. In vivo research models include animal experiments, gene knockout animal models, positron emission tomography (PET) technology, and studies on human beings. In addition, in vitro–in vivo extrapolation (IVIVE), ex vivo kidney perfusion (EVKP) models, and, more recently, biomarker methods and in silico models are included. This article reviews the traditional research methods of renal transporter-mediated DDIs, updates the recent progress in the development of the methods, and then classifies and summarizes the advantages and disadvantages of each method. Through the sorting work conducted in this paper, it will be convenient for researchers at different learning stages to choose the best method for their own research based on their own subject’s situation when they are going to study DDIs mediated by renal transporters.

## 1. Introduction

Drug–drug interactions (DDIs) are a key issue in clinical rational administration and post-marketing pharmacovigilance. The presence of additional drugs may significantly alert the activity of one drug when taking two or more drugs concurrently or consecutively, a condition known as a DDI [1]. With the change in the modern disease spectrum and the increase in drug resistance, drug combinations are often needed in clinical practice. Combinations of drugs may increase the risk of clinically relevant DDIs and pose new challenges for treatment management [2]. In addition to the well-known CYP450 enzyme family and two-phase metabolic enzymes, which are important targets for the occurrence of DDIs, transporter-mediated DDIs have also been paid more and more attention by researchers [3,4]. Transporters affect the processes of drugs in the body, resulting in the accumulation or excretion of drugs, thereby enhancing their pharmacological effects or producing side effects. For this, regulatory agencies, such as the Food and Drug Administration (FDA), have issued new recommendations for the assessment of potential risks associated with DDIs in the development of new drugs [5].

The kidney is essential for preserving body homeostasis and removing harmful compounds. Eventually, numerous drugs and their metabolites are eliminated in urine [6]. Impaired renal function can reduce drug excretion and metabolism significantly. This may result in an increase in the blood concentrations of drugs, which can affect their efficacy and toxicity. Transporters, which are a type of membrane protein, can be divided into two main superfamilies: solute carrier transporters (SLCs) and ATP-binding cassette transporters (ABC). ABC transporters, which include roughly 50 members separated into 7 families, mediate drug efflux from cells, whereas SLC transporters, which have over 400 membrane proteins assigned to over 60 families, participate in drug absorption processes [7]. Transporters can affect the therapeutic effects, side effects, and DDIs of drugs. In other words, they are crucial in drug pharmacokinetics and pharmacodynamics [8]. There has been considerable research over the past few decades showing that renal transporters affect renal excretion, renal toxicity, and DDIs [6].

Renal transporter-mediated DDIs can be assessed in vitro, in vivo, in vitro-in vivo extrapolation (IVIVE), and ex vivo kidney perfusion (EVKP) models, etc. Recently, new research methods, such as biomarkers and in silico models, have also become suitable for predicting renal transporter-mediated DDIs. A study may need the combined use of multiple methods to investigate renal transporter-mediated DDIs. Therefore, it is important to know which model should be applied properly at different research stages and how the model can be most accurately and efficiently utilized. This article reviews the basic and the latest research methods for DDIs mediated by renal transporters, and further discusses the advantages and disadvantages of these methods so as to facilitate the research of scholars at different learning stages [9].

## 2. Overview of Kidney Transporters

Drug transporters in the human body can be divided into two categories according to their functions. One is the SLC superfamily that mediates drug absorption into cells, which has a broad distribution across the human body’s tissues and organs. Another type of transporter is the ABC superfamily, which can utilize energy by hydrolyzing ATP to transport drugs out of cells, thereby reducing the intracellular drug concentration and producing drug resistance [10]. In different organs of the body, transporters have different distributions and expressions, and are mostly expressed in drug disposal organs to promote the influx and efflux of exogenous compounds and endogenous substrates [11]. In the kidney, organic cation transporters (OCTs), organic anion transporters (OATs), and multidrug and toxin efflux proteins (MATEs), SLC family members, are more abundant than P-glycoprotein (P-gp), multidrug resistance-associated proteins (MRPs), and breast cancer-resistance protein (BCRP), ABC family members. OAT1, OAT3, OCT2, and MATE1 are richly expressed vectors in SLCs, while P-gp, MRP2, and MRP4 are the most expressed vectors in ABC [12]. They are shown in Figure 1.

### 2.1. The Superfamily of SLC

The uptake transporters of the SLC superfamily are secondary active transporters, which are based on the physicochemical properties of the substrate and membrane localization of different transporters, and rely on the driving forces of potential difference and ion concentration gradient on the cell membrane [9]. The SLC family is essential for transporting diverse ions and organic compounds in the renal tubule. Furthermore, certain members of the SLC family facilitate essential metabolic processes by transporting substrates required for metabolism [13]. Renal SLC transporters mainly include: (1) OATs, (2) organic anion-transporting polypeptides (OATPs), (3) OCTs, (4) MATEs, and (5) peptide transporters (PEPTs).

#### 2.1.1. OATs

OATs are products of the SLC22 gene family. OAT1, OAT3, and OAT4 are primarily found in the kidneys, while OAT2 is largely present in the liver. The expression patterns of OAT1, OAT3, and OAT4 differ in renal proximal tubular cells. OAT1 and OAT3 are primarily expressed at the basolateral membrane, whereas OAT4 is mainly located at the apical membrane [4,14]. The substrate specificities of OAT1 and OAT3 overlap significantly. However, OAT3 has a larger affinity for lipophilic organic anions [15]. Among medicinal drugs, OAT substrates include β-lactam antibiotics, methotrexate, antiviral drugs, diuretics, etc. [16]. These substrates make it clear that one of the main characteristics of OATs is their interaction with and transportation of a wide range of molecules with dissimilar chemical structures. The substrates primarily require a negative charge and a hydrophobic region, and there are multiple points of interaction between the carrier and functional groups of the substrate, particularly the carbonyl and carboxyl groups. As OATs can recognize multiple substrates, DDIs can occur at their transport sites. When multiple drugs coexist in the plasma, they can compete for binding to the transporter, thereby mutually impacting their pharmacokinetics [15].

#### 2.1.2. OATPs

OATPs are products of the SLC21 gene family. These OATPs are widely distributed in organs and tissues closely related to drug absorption and disposal, such as the liver, placenta, brain, and kidney. Owing to their wide tissue distribution and substrate selectivity, multiple drugs can interact at multiple stages of absorption and disposal [17]. OATP4C1 is the only OATP family transporter expressed in the kidney and localized at the basolateral membrane of the proximal tubules [16]. OATPs are equipped with multiple sites for binding substrates or pathways for translocation. For instance, OATP4C1 was found to possess different binding sites for estrone-3-sulfate and digoxin. Furthermore, studies on inhibition have revealed that compounds can elicit a stimulatory, inhibitory, or no effect on OATP-mediated transport, which varies depending on the specific model substrate being examined [18].

#### 2.1.3. OCTs

Renal OCTs mainly include OCT1, OCT2, and OCT3, among which OCT1 is predominantly expressed at the apical membrane of renal proximal tubular cells, while OCT2 and OCT3 are mainly expressed at the basal membrane of these cells. OCTs participate in the renal excretion of most cationic drugs, endogenous organic cations, and toxins in vivo, as well as in the reabsorption of some endogenous substances and drugs after glomerular filtration. Therefore, the pharmacokinetic processes and nephrotoxicity of cationic drugs are often closely related to the function of renal OCTs [19,20,21,22]. Complex cation-binding regions are present in OCTs. High-affinity cation binding sites have the potential to cause the allosteric inhibition of transport, whereas overlapping low-affinity cation binding sites are directly implicated in transport. Remarkably, high-affinity inhibition is only seen when absorption is assessed using substrate concentrations at nanomolar levels that are much lower than the corresponding Km values. The affinities of inhibitors depend on their molecular makeup and substrate concentration [23].

#### 2.1.4. MATEs

MATE1 and MATE2-K among the MATEs are mainly expressed at the apical membrane of renal proximal tubular cells and mediate the final excretion of organic cationic drugs and metabolites into urine through proton exchange [24]. The human MATE proteins are capable of transporting various substances, including creatinine, corticosteroids, metformin, and cimetidine, as well as certain antibiotics. Cimetidine, levofloxacin, and pyrimethamine are potent inhibitors of MATE transporters [16]. The drug-binding sites of MATE proteins are usually located in the spatial region between the transmembrane and cytoplasmic domains. By binding to drugs, MATE proteins can influence drug transport and excretion. When multiple drugs are present together, they may compete to bind to the drug-binding sites of MATE proteins, thereby affecting each other’s pharmacokinetics [25].

#### 2.1.5. PEPTs

PEPTs transport dipeptides, tripeptides, and peptide analogs with the assistance of the Na^+^/H^+^ exchange system using the H^+^ gradient as a transmembrane driver against the concentration gradient. PEPT1, a low-affinity peptide transporter, is primarily expressed in the small intestine and is also found in low amounts in the kidneys. On the other hand, PEPT2, a high-affinity peptide transporter, is mainly expressed at the apical membrane of renal proximal tubular cells [26,27,28]. The structure of PEPTs contains multiple binding sites for binding to peptides. These binding sites typically include amino acid residues, such as aromatic amino acids (e.g., phenylalanine and tyrosine) and positively charged amino acids (e.g., arginine and lysine), which interact with the chemical structure of the peptide. Drugs can bind competitively to these binding sites, thereby affecting the binding and transport of the peptide [29].

### 2.2. The Superfamily of ABC

The efflux transporters from the ABC superfamily are mediated by ATP-dependent primary active transport. They can utilize energy by hydrolyzing ATP to transport drugs out of cells, thereby reducing the intracellular drug concentration and producing drug resistance [30]. Renal ABC transporters mainly include: (1) P-gp, (2) MRPs, and (3) BCRP [31].

#### 2.2.1. P-gp

P-gp was the first ABC transporter discovered in human tissue and also one of the most characteristic [32]. In the kidneys, P-gp is mainly expressed at the apical membrane of renal proximal tubular cells. P-gp has a very extensive range of substrates, mainly hydrophobic cationic compounds [33]. The decrease in digoxin clearance observed with quinidine, verapamil, ritonavir, and itraconazole is assumed to result from P-gp interactions in the kidney [16]. According to common opinion, P-gp has at least three drug/substrate-binding sites and one allosteric site; they are separate, but interact with one another to perform the transporter function of P-gp [34]. Many known P-gp inhibitors are competitive inhibitors because they preferentially interact reversibly with one or more of the three hypothesized drug/substrate-binding sites [35]. 

#### 2.2.2. MRPs

In the kidneys, MRP2 and MRP4 are mainly expressed at the apical membrane of proximal tubule cells, both of which actively transport substrate drugs and metabolites out of cells, whereas MRP3 is localized at the basolateral membrane of the distal tubule [28,36]. A variety of conjugated drug metabolites, as well as bilirubin–glucuronide and leukotriene C4, are among the substances that are transported by MRPs [9]. It is believed that MRP1 directly recruits its substrates from the cytoplasm. Membrane partitioning of the substrate is not necessary for MRP1 to recognize various substrates. Instead, it comes from its single substrate-binding site’s bipartite nature and plasticity. MRP1 may recognize a variety of substrates with various chemical structures by establishing a single bipartite substrate-binding site [37].

#### 2.2.3. BCRP

BCRP is the only half-transporter in the ABC transporter family. In II-phase binding, it plays a crucial role in the efflux transport of medicines. Renal BCRP is mainly expressed at the apical membrane of proximal tubule cells and plays a crucial role in the excretion of organic cations in the kidney [38]. The main substrates of BCRP include endogenous substance urate and clinical drugs, such as cimetidine, imatinib, topotecan, irinotecan, methotrexate, nilotinib, prazosin, rosuvastatin, etc. [31]. BCRP contains numerous drug-binding sites that differ in terms of location within the binding pocket and/or affinity [39].

## 3. Effect of Kidney Disease on Renal Drug Transporters

It is apparent that during kidney pathology, there is dysregulation of transporters [31]. Kidney diseases are known to exert an impact on the expression and function of drug transporters in the organ. Table 1 shows a summary of transporter expression changes across kidney diseases. To better define drug pharmacokinetics and DDIs, as well as to comprehend the organ pathophysiology, it is crucial to understand how kidney illnesses affect drug transporter expression [40].

In a cell model, Naud J. et al. [41] discovered that HK-2 cells exposed to sera from rats with chronic renal failure (CRF) significantly upregulated the protein expression levels of MRP2/3/4 and OATP2/3, while notably downregulating the levels of P-gp, OAT1/2/3, and OATP1/4C1. These results are consistent with research on animals that had CRF caused by experimentation. Matsuzaki T et al. [42] observed that the levels of mRNA and protein of both OAT1 and OAT3 were noticeably reduced in the ischemic kidney during acute renal failure (ARF) generated by the ischemia/reperfusion of rat kidneys. In addition, uric acid nephropathy is frequently associated with decreased OAT1/3 mRNA and protein levels [43,44]. Clinical studies [45] revealed significantly lower OAT1 mRNA expression in kidney biopsy samples from IgA nephropathy patients. In summary, the expression of transporters in vivo changes to different degrees during the disease process and drug interaction, suggesting that transporters may be used as new markers for clinical disease diagnosis and prognosis. With the in-depth study of clinical drug interactions, numerous studies have shown that drug combinations can reduce toxicity and enhance efficacy by regulating the expression and function of transporters. This provides a strong basis for clinically rational drug use.

Renal damage caused by various chronic kidney diseases and systemic chronic diseases will eventually evolve into CRF. CRF will not only reduce the glomerular filtration rate, but also affect the activity of drug metabolism enzymes and transporters. The changes in transporters may affect the processing of drugs in vivo, resulting in an increase or decrease in the blood drug concentration. At the same time, it also mediates the occurrence of DDIs, which, in turn, affects the efficacy of drugs. In severe cases, it can lead to adverse reactions and even endanger the lives of patients [46,47]. Therefore, it is necessary to study the effect of kidney disease on renal drug transporters.

**Table 1 molecules-28-05252-t001:** Summary of transporter expression changes across kidney diseases.

Disease		Transporters	Expression Level Compared with Healthy	References
Chronic Kidney Disease	Diabetic Nephropathy	MRP1	Increased	[48]
		P-gp, PEPT1, PEPT2	Increased	[49]
	Immune Nephropathy	OAT1	Decreased	[45]
	Chronic Renal Failure	P-gp, OAT1/2/3, OATP1/4C1	Decreased	[41,50]
		MRP2/3/4, OATP2/3	Increased	
		OATP4C1, MATE1, PEPT1, OAT1/3, OCT1/2	Decreased	[51]
		P-gp	Increased	
		PEPT2	Increased	[52]
		OCT2	Decreased	[53]
Acute Kidney Injury		OCT2, MATE1	Decreased	[54]
		OCT1, OCT2	Decreased	[55]
		OAT1, OAT3	Decreased	[42,56,57]
		MRP2	Increased	[58]
		OAT1, OCT2, OCT3	Decreased	[59]

## 4. Renal Transporter-Mediated DDIs

Multidrug regimens in the current prevalence of disease conditions have led to a vast range of DDIs. The induction or inhibition of one drug on the processes involved in the transport of another drug can lead to DDI. The kidneys are one of the most important target and excretion organs in the body. Transporters are widely expressed in the kidneys and play a key role in the secretion and reabsorption of many endogenous and exogenous substances. Thus, the study of renal transporter-mediated DDIs has attracted much attention [60].

Metformin [61] is a commonly used first-line oral hypoglycemic drug in clinical practice for treating type 2 diabetes mellitus. As type 2 diabetes can be complicated by cardiovascular, ocular, neurological, and renal pathologies, co-treatment with multiple drugs is often required. Metformin is primarily a substrate for OCT and MATE transporters, and cimetidine is a renal OCT transporter inhibitor. When metformin is combined with cimetidine, cimetidine significantly inhibits the transport of metformin from the basolateral to the apical membrane side, and DDI occurs, which can lead to lactic acidosis and acute kidney injury associated with acute pancreatitis in patients with type 2 diabetes. Methotrexate [31] (MTX) is actively excreted via the kidney, and its action is mainly mediated by a combination of uptake transporters (OAT1 and OAT3) and efflux transporters (MRPs and BCRP). Non-steroidal anti-inflammatory drugs have inhibitory effects on OATs, leading to the adverse impaired renal elimination of MTX, drug accumulation, and severe bone marrow suppression effects. In addition, examples of DDIs mediated by OCT2 transporters in the kidney include metformin and pyrimethamine [62], cisplatin and vandetanib [63], dofetilide and cimetidine, etc. [64]. Examples of DDIs mediated by OAT1 transporters include acyclovir/zalcitabine and probenecid [65,66]; DDIs mediated by P-gp transporters include digoxin, verapamil, etc. [67]. Examples of classical DDIs mediated by renal transporters are listed in Table 2. More examples of renal transporter-mediated DDIs can be found in this excellent review [16].

## 5. Methods for the Study of DDIs Mediated by Renal Transporters

Since the 1980s, with the development of molecular biology, the study of renal transporters has made rapid progress. The exploration of these transporters has helped to improve drug safety and efficacy, played an important role in understanding drug toxicity and DDIs, and also provided a theoretical basis for improving drug targeting. Regarding renal transporters, researchers and drug discovery scientists have studied a lot in the field of their mediated DDIs, from traditional models to recent biomarker methods and in silico models. Scholars performing pharmacokinetics work are still working on it, which is convenient for beginners and experienced scholars to consult and learn. The current common research methods can be sorted out as follows, as shown in Figure 2.

### 5.1. In Vitro Research Models

In the early stages of drug research, in vitro models are critical in establishing whether a drug candidate is a transporter substrate or inhibitor. In vitro screening of drug–transporter interactions helps to predict the susceptibility of DDI mediated by transporters in vivo. In vitro models provide information related to the kinetic parameters of substrates (Km and Vmax) and inhibitors (Ki, IC50, or Vmax) for analyzing potential drug interactions [60]. Methods for in vitro studies of renal transporter-mediated DDIs mainly include membrane-based assays, cell-based assays, and renal slice uptake assays.

#### 5.1.1. Membrane-Based Assay Systems

##### Membrane Vesicle Transport Assay

The membrane vesicle transport assay was the first membrane-based assay widely used in the study of the ABC superfamily. Membrane vesicles are formed by separating the plasma membrane from the cell expressing the transporter protein and providing ideal conditions (pH, temperature, and cofactors) for the preparation of membranes containing inside-out vesicles, ATP binding sites, and substrate-binding sites toward the outer buffer. If a drug accumulates within the vesicles in an ATP-dependent and concentration-dependent manner, it indicates that the transporter under investigation is engaged in its transport. If, on the other hand, a medicine inhibits the accumulation of the probe substrate, it is deemed a transporter inhibitor [60,79].

The membrane vesicle transport assay can be easily adapted to a multi-well plate format, making it suitable for high-throughput analysis. In addition, the use of commercially available vesicles can eliminate the variability between laboratories and preparations as they are produced in large quantities and can be stored at either −80 °C or in liquid nitrogen while maintaining their activity. Despite its advantages, the vesicle assay has a significant restriction when employed for hydrophobic substrates, as it can lead to false-negative results due to non-specific binding and vesicle leaking. However, this limitation can be minimized by using a rational study design, such as consistent procedures, consideration of compound solubility, and an appropriate incubation time [80]. Deng, F et al. [81] tested 232 drugs using membrane vesicle transport assays and found that many of the drugs already on the market were inhibitors of BCRP.

##### ATPase Assay

The early stage of development frequently employs ATPase for assessing efflux transporter interactions and screening DDIs. The colorimetric approach can be used to measure inorganic phosphate from ATP hydrolysis and reflect the transport activity [60,82]. For example, Satoh, T. et al. [83] used a human P-gp membrane ATPase assay in order to research the impact of Kampo medicines on P-gp and the DDIs between Kampo drugs and western pharmaceuticals. The study found that the majority of Kampo medicines inhibited ATPase activity, suggesting that they might inhibit P-gp function.

As the ATPase assay does not analyze medicines using radioactivity or specialized equipment, it is appropriate for high-throughput screening and allows for the batch analysis of compounds that interact with ABC transporters. However, the ATPase assay has some limitations. Firstly, the assay cannot directly identify transporter substrates or inhibitors due to its indirect measurement of transport. Second, some substrates and inhibitors’ ATPase activities do not match their transport rates, which can lead to results that are either falsely positive or falsely negative. Furthermore, a larger concentration of substrate is required, etc. [82].

#### 5.1.2. Cell-Based Assays

##### Primary Cells

Primary cells are produced from intact tissues and are capable of expressing all transporter genes found in that tissue. They can be used to study drug metabolism, transport, and clinical drug interactions. Janneh, O. et al. [84] assessed DDIs at the level of drug transport using peripheral blood mononuclear cells (PBMCs). The researchers used flow cytometry to measure the expression of P-gp, MRP1, and BCRP in PBMCs. However, although P-gp, MRP1, and BCRP were detected in PBMCs, there was no observed correlation between their expression and drug accumulation. As a result, an interaction at the transporter level does not fully account for the high failure rates observed with drugs such as tenofovir, abacavir, and [3H]-lamivudine. Lash, L.H. et al. [85] discovered that primary cultures of human proximal tubular cells expressed a varied array of transporters for important classes of significant drugs and were useful for investigating drug transport and disposal, as well as assessing potential DDIs in the human kidneys.

When characterizing transporter activities, using primary cells offers various benefits. The primary cells have an intact cellular architecture, a functional membrane, cytoplasmic elements, and cotransporting ions, which can lead to more reliable results. In addition, the primary cells are derived from intact tissues and can express all transporter genes present in that tissue. However, several unique transporters are commonly not expressed in primary cells, and transport tests are often performed in stable cell lines or transfected cells expressing the special transporters [82].

##### Transfected Cells

There are numerous cell lines used to construct transfected cells. As transporter-transfected cells, MDCKII, LLC-PK1, Chinese hamster ovary (CHO), HEK293, and HeLa cells have been extensively used. The construction process includes the transfection of recombinant plasmids containing the target gene fragments into specific cells, screening with G418, the selection of monoclonal cells, and the verification of the successful construction of transfected cells by combining Western blot, PCR, immunofluorescence microscopy, the radiolabeled substrate uptake/exclusion rate, etc.

Many single-transfected cells, such as OAT1/3-HEK293 [86], OCT2-HEK293 [87], PEPT2-HELA [88], and BCRP/MDR1-MDCK [89], have been successfully established. In a study, ranitidine absorption and its inhibitory effects on other medicines were examined in HEK293 or CHO cells that had been stably transfected with OCT1, OCT2, or their allelic variations. The results showed that ranitidine had the potential to cause DDIs when coadministered with OCT1 substrates and that OCT1 genetic variations significantly affected ranitidine absorption [90]. In addition, the fundamental mechanism of renal DDIs can also be understood using double-transfected cell lines [60]. Although it is difficult to construct double-transfected cells, MDCK-OCT1/2-MATE1 [91] and MDCK-OATP4C1-P-gp [92] cell models have been successfully established. König et al. [91] constructed MDCK-OCT1-MATE1 and MDCK-OCT2-MATE1 double-transfected cell models and used the corresponding single-transporter transfected cells as controls to research the transport mechanisms of metformin and the methyl–phenyl–pyridine cation, and confirmed that OCT1 and OCT2 mediated the uptake transport of both drugs, while MATE1 mediated the efflux. George, B et al. [93] investigated MDCK cells by double-transfecting OCT2-MATE1 and discovered that 5-HT3 antagonist drugs had the potential to inhibit the renal secretion of cationic drugs by interfering with the function of OCT2 and/or MATE1. Müller et al. [94] used double-transfected MDCK-OCT2-MATE1 cells as a model to simulate the organic cation transport processes in proximal renal tubule cells and to investigate the significance of OCT2 and MATEs in the interaction between cimetidine and metformin. The above approach allows not only to examine the transporters that mediate the transport of each substrate, but also to study the interactions of multiple transporters.

It is generally believed that single-transfected cells usually lack endogenous uptake or efflux transporters and cannot mimic the complete mechanism of the transmembrane transport of drug molecules, while double-transfected cells can overcome this defect to a certain extent. However, in intact organs, the transport of certain compounds may be mediated by multiple transporters, and even double-transfected cells cannot fully predict the true process of in vivo transport, so some researchers have constructed triple-transfected cells. For example, Hirouchi et al. [95] constructed MRP2/MRP3/OATP1B1 and MRP2/MRP4/OATP1B1 triple-transfected cells for studying the vectorial transport of drugs through the transporter.

Using transfected cells can cause any type of cell to express the transporter of interest, even if the transporter is not expressed in that type of cell. Moreover, the expression of transporters in cells can be controlled to a certain level. However, it is worth noting that the transfection process may change the cellular environment in which the transporter is located, which may affect the true expression and function of the transporter.

##### Three-Dimensional (3D) Cells

Conventional cell culture does not adequately simulate the in vivo environment; therefore, its use as a model for relevant studies can be biased or affect the use of experimental data. Unlike conventional cell cultures, 3D cell cultures can better simulate the natural environment in which cells survive in an organism. Even simple spherical models can compensate for many of the shortcomings of monolayer cultures. These structures can provide oxygen, nutrition, and metabolites, resulting in different cell populations. However, there are some disadvantages to using 3D cells, such as the fact that 3D cell culture consumes more time and resources and is more costly. Additionally, the research process is more complex and requires more technical and equipment support [96,97,98].

There are many 3D cell culture technologies, which can be summarized into three categories: 3D hydrogels, cell aggregation, and culture scaffolds. As hydrogels have biophysical properties very similar to those of natural tissues, they can be used as efficient 3D cell culture matrices. Examples of hydrogel technologies are natural hydrogels, extracellular matrices, etc. Cell aggregation is the phenomenon of several cells coming together to form clusters. Cell aggregation is related to the physiological characteristics of the cells, especially the material structure of the cell wall, with technical examples such as suspension drop culture plates, low-adherence planes, etc. Culture scaffolds provide a physical support into which cells can enter to grow and perform their functions; technical examples are natural scaffolds (collagen and gelatin) and synthetic scaffolds (polystyrene, polyurethane, etc.) [99]. Vriend, J. et al. [100] developed a 3D microfluidic proximal renal tubular epithelial cell model and demonstrated the functional activity and drug interactions of P-gp and MRP2 and 4 by fluorescence-based transport assays. This model proved to be suitable for assessing the interaction of renal drugs with efflux transport proteins.

#### 5.1.3. Renal Slice Uptake Model

In the renal slice uptake model, the kidneys of anesthetized rats are removed and cut into slices of approximately 300 μm thickness. The slices are incubated in an oxygenated water bath, the drug to be tested is added, and the amount of drug taken up is measured. Renal slice uptake assays are often used to examine whether the drug is a substrate or inhibitor of OAT1/3 and OCT, to predict the renal transport characteristics of the compound, and to make predictions for later transfection cell experiments. The benefit of this model is that studies using human or animal kidney tissue can more closely resemble the in vivo environment.

For example, Yang, S. et al. examined the uptake of piperacillin and tazobactam using renal slices. The findings revealed that rOat1 and rOat3 mediated a beneficial interaction between the two drugs [101]. Zhang et al. conducted pharmacokinetic studies and uptake assays using rat renal slices and hOAT1/3-HEK293 cells to assess the potential DDI between bentysrepinine and entecavir. The study results led to the conclusion that it is safe to use bentysrepinine with entecavir in clinical practice [102]. Xu, Q. et al. utilized LC-MS/MS to determine the plasma and urine concentrations of entecavir after intravenous and oral administration in vivo, as well as assess entecavir uptake in kidney slices and transfected cells in vitro. The study findings demonstrated that OAT1 and OAT3 are DDI targets between entecavir and JBP485 (a dipeptide) [103]. The above studies on renal transporter-mediated DDIs all involved renal slice uptake assays, but they are not applicable to all renal transporter studies. Owing to the lack of a complete renal tubular network, renal slice uptake assays can only examine the uptake of drugs by renal secretory transporters, and not reabsorption transporters. In addition, it is worth noting that the cutting machine may damage the tissue structure, resulting in large errors in the experimental results.

In vitro models still have suboptimal predictive performance for clinically relevant drug interactions, despite being widely used, because these in vitro models do not allow for the investigation of the coordinated action of all transporters that occurs in real-life epithelial cells. Furthermore, they frequently overlook the effects of drug metabolites, which cannot form in vitro [16]. 

### 5.2. In Vivo Research Modelsd

#### 5.2.1. Animal Experiments

Rodents (rats and mice) and non-rodents (monkeys) are the two types of animals that are most frequently employed in transporter-mediated DDI research [104]. Rodents, such as Sprague–Dawley rats, Wistar rats, and wild-type mice, have been widely utilized as the first species for pharmacokinetic studies because they are affordable, have good reproductive performance, and are resistant to infectious diseases [60]. To investigate whether probenecid can alter the in vivo pharmacokinetics of biapenem, Li, W. et al. dissolved biapenem and probenecid in saline and injected it intravenously through Sprague–Dawley rats’ tail veins and then collected plasma and urine samples for study. The researchers concluded that renal tubular secretion mediated by OAT3 is a minor pathway for biapenem clearance. Biapenem would be safe to use in conjunction with other antibiotics and antiviral medicines in a clinical environment [105].

The use of animal experiments allows for a more comprehensive understanding of drug metabolism and transport in vivo. By comparing differences between species, the properties of transporters can be further investigated. However, the huge disparities between native animal models and humans in terms of transporters make it challenging to interpret the results. New preclinical models, such as gene knockout animal models, have been developed as a result of the growth of molecular biology and engineering tools.

#### 5.2.2. Gene Knockout Animal Models

Schinkel et al. [106] used gene editing technology to construct mdr1a (−/−) mice, which have no obvious physiological defects, but do not express P-gp. Subsequently, there have been increasing types of gene knockout animal models with improved commercial applications, and a series of gene knockout mice related to drug transporters have emerged, such as BCRP (−/−) mice, MRP2 (−/−) mice, MRP4 (−/−) mice, PEPT1 (−/−) mice and OCT1 (−/−) mice, and so on [82]. Gene knockout animal models have become one of the most important tools for studying transporter functions today. In one study, Breedveld, P. et al. [107] used BCRP (−/−) mice and wild-type mice to assess the mechanism of interaction of benzimidazole with MTX. The conclusion was as follows: the clinical interaction between MTX and benzimidazole may be explained by competition for BCRP. In addition to this, Kikuchi, R. et al. [108] used OCT1/OCT2 double-knockout mice to study the clinical drug interactions of veliparib with renal transporters and found that OAT1/3-, OCT2-, and MATE1/2K-mediated DDIs were the least likely.

Gene knockout animal models are of great significance for the evaluation of renal transporter-mediated DDIs, as well as drug disposition. The use of gene knockout animal models has the following benefits: ① they are closer to the physiological mechanisms of the human body; ② transporter function can be elucidated under physiological conditions; and ③ drug uptake and efflux by transporters can be studied without inhibitors. However, when using these models to understand transport functions, care should be taken, because the deletion of one transporter can often cause particular alterations in the expression pattern and function of other transporters, as well as alter the physiology of animals. For example, in MRP2-deficient rats, the expression of the MRP3 protein was greatly activated [109]. In addition, the limited variety, high cost, and interspecies differences in gene knockout animal models make it difficult to apply this technology for the high-throughput screening of transporter substrates and inhibitors [110].

#### 5.2.3. Positron Emission Tomography (PET) Technology

PET is a non-invasive imaging technique that can study the distribution of radiolabeled medicines in various organs and tissues. Moreover, it allows for the real-time monitoring of drug metabolism and transport processes. Therefore, PET is the preferred method for quantitatively evaluating transporter-mediated DDIs at the tissue level. PET has been applied in both preclinical animal models and human subjects to evaluate the impacts of transporter-mediated DDIs on drug disposition across various organ systems, including the brain, liver, and kidneys [111]. For example, Hernandez-Lozano, I. et al. [112] evaluated the impact of different drugs, which may result in transporter-mediated DDIs, on the tissue distribution and excretion of ciprofloxacin. Additionally, they used PET imaging-based pharmacokinetic (PK) analysis. The outcomes demonstrated that DDIs can occur due to the inhibition of renal transporters by concomitant drugs, resulting in decreased urinary excretion and increased blood and organ exposure to ciprofloxacin. Another study [113] conducted in rats and pigs used ^11^C-metformin PET to evaluate the function of OCT transporters in the kidneys and liver. The studies both emphasized the advantages of PET imaging-based PK analysis to evaluate transporter-mediated DDIs at a whole-body level. However, it is worth noting that PET technology also has some disadvantages, such as the fact that it requires the use of radioisotope-labeled drugs and has radiological safety issues, as well as expensive equipment, requiring a high level of technical support and operators.

In vivo PET imaging can accurately quantify the impact of transporter-mediated DDIs on the distribution of radiolabeled drugs in various organs and tissues. This novel method shows tremendous potential for elucidating the role of transporters in drug distribution and may prove helpful in the creation of new drugs. Continued research is being conducted to identify new PET radiotracers that target specific transporters, which is expected to expand the range of applications for PET in transporter studies [111,114].

#### 5.2.4. Human Beings

Although a lot of results have been obtained from in vitro studies or animal experiments in the study of DDIs, it is often difficult to provide clinical guidance or application. The reason is that many results are conflicting, and the interactions that occur in vitro or in animal bodies may not necessarily occur in the human body [115,116]. Therefore, under the ethical guidelines of medical research involving human participants, research on healthy volunteers has gradually been carried out. For example, Arun et al. [117] evaluated the potential DDIs between sitagliptin and gemfibrozil in a study of 12 healthy Indian male volunteers. The researchers concluded that gemfibrozil significantly elevated the AUC0-∞ of sitagliptin, possibly by inhibiting hOAT3 transporters at renal tubules. In a study conducted by Morrissey, K. M. et al. [118], the impact of nizatidine on the pharmacokinetics and pharmacodynamics of metformin was investigated in 12 healthy volunteers. The results showed that nizatidine had no effect on metformin systemic concentrations or CLr, indicating that specific MATE2K inhibition may not be enough to generate renal DDIs with metformin.

This model can be used to directly understand the metabolism and operation of drugs and their metabolites in the human body. However, it is essential to note that various factors, such as gender, age, social habits, disease conditions, and genetic predisposition, among others, play a major role in DDIs, resulting in great interindividual variability. To take these factors into account, future research studies should involve large patient populations [117].

### 5.3. In Vitro–In Vivo Extrapolation (IVIVE)

The IVIVE method essentially uses preclinical in vitro and in vivo data to find correlations and then conducts corresponding in vitro studies in humans based on this correlation in order to predict the CLr in the human body. Before human dosing or prioritizing specific types of clinical DDI studies, in vitro data need to be integrated, understood, and translated to support risk assessment. To date, there have been few examples of IVIVE involving renal transporters for DDIs. Shen, H et al. [119] discovered that the cynomolgus monkey may be useful in the support of IVIVEs that involve the inhibition of renal MATEs and OCT2. In addition, IVIVEs adapted from cynomolgus monkeys may provide insight into the risk of DDI in humans.

With the advancement of IVIVE technology, its applications are expanding rapidly, extending from the original animal species to the human species. IVIVE is seen as a potential method for forecasting transporter-mediated drug CLr or DDIs in humans. It can reduce the need for people to experiment on animals and reduce the burden caused by animal ethical issues. However, transporter-based IVIVE in humans is still in an early stage due to the following reasons: (1) Without specific substrates or inhibitors, it is hard to identify the contribution of different transporters. The involvement of numerous transporters with overlapping inhibitors may undervalue or overvalue the contribution of transporters in drug disposition. (2) Some transporters may have multiple drug-binding sites, which can further complicate predictions. (3) There is still limited knowledge on species differences in drug transporters, which also adds to the complexity of IVIVE. (4) It needs a lot of data. In a word, caution should be taken when extrapolating in vitro data to an in vivo setting [9]. 

### 5.4. Ex Vivo Kidney Perfusion (EVKP) Models

Ex vivo perfusion (EVP) models, such as ex vivo perfused lung, intestine, brain, liver, and kidneys, are commonly employed in drug transport research. The process of perfusion may require complex equipment and technical support. These models allow for a more physiological environment to determine transporter functionality compared with in vitro experiments. Additionally, EVP models avoid the potential confounding effects of other organs on drug disposal that may occur in in vivo studies. EVP models examine the absorption or efflux role of transporters on drugs by adding selective inhibitors of transporters to the perfusate and comparing the differences in drug levels in the perfusate, tissues, organs, or plasma. By comparing the results of single- or multiple-drug combination perfusion, the competition of the same transporter by the combined drug can be studied [120,121].

The EVKP model can be used to study the metabolism and excretion of the kidneys. For example, Posma, R. A. et al. [122] investigated whether adding metformin prior to or during ex vivo isolated normothermic machine perfusion (NMP) of pig and rat kidneys affected its elimination. The metformin CLr was significantly greater than creatinine CLr, confirming metformin production during the ex vivo NMP of both rat and porcine kidneys. In addition to elucidating the CLr mechanism, the EVKP model can be used to predict DDIs. For example, Hori, R. et al. [123] utilized an isolated perfused rat kidney model to study the renal tubular secretion mechanism of digoxin and its interaction with quinidine or verapamil. Their findings suggested that digoxin is a substrate transported by P-gp and that P-gp inhibition causes clinically significant interactions with quinidine and verapamil.

### 5.5. Biomarker Methods

Endogenous biomarkers, which are generally physiological substrates of drug-metabolizing enzymes and transporters, have recently emerged as effective tools for advancing the risk assessment of DDIs [124]. A number of biomarkers predicting renal transporter activity have been identified, including 6β hydroxycortisol (6β-OHF), taurine and glycochenodeoxycholate sulfate (GCDCA-S), thiamine, N-methylnicotinamide (NMN), creatinine, N1-methyladenosine (m^1^A), and so on.

#### 5.5.1. Kidney Endogenous Biomarkers of OAT1 and OAT3

6β-OHF is a prominent endogenous in vivo probe that is commonly used to evaluate the inhibition of OAT3. It is synthesized by hepatic CYP3A4 and eliminated from the body through urine [125]. Imamura et al. used transporter-expressing cell lines to determine that 6β-OHF is a substrate of MATE1, MATE2-K, and OAT3 [126]. In another study, Imamura et al. used in vivo inhibitors, probenecid and pyrimethamine, to selectively inhibit OAT3 and MATEs, respectively, and investigate the role of OAT3 and MATEs in the urinary excretion of 6β-OHF in humans. The in vivo and in vitro results indicated that OAT3 significantly contributed to the urinary excretion of 6β-OHF and that 6β-OHF can be utilized to evaluate OAT3-mediated drug interactions in humans [127].

Taurine is an amino acid that can be acquired from the diet or be produced in the body, whereas GCDCA-S is a significant sulfated bile acid conjugate present in plasma and urine. Taurine and GCDCA-S have been suggested as endogenous probes for assessing the inhibition of OAT1 and OAT3, respectively [128]. Tsuruya et al. [129] investigated the effect of probenecid on the alterations in endogenous chemicals in plasma and urine samples using metabolomics analysis and found that taurine and GCDCA-S can be utilized as probes to assess pharmacokinetic DDIs involving OAT1 and OAT3, respectively, in humans.

#### 5.5.2. Kidney Endogenous Biomarkers of OCT2 and MATE1/2K

Thiamine [130], NMN [131,132], and creatinine [133] have been shown in previous studies to be potential endogenous biomarkers of OCT2 and MATE1/2K in the kidneys. Among them, creatinine is commonly utilized as a biomarker for renal function. Several compounds were positively predicted to result in OCT2/MATEs-mediated DDIs based on in vitro–in vivo correlation studies between the terms of inhibition of OCT2 and MATE1/2K and clinically observed changes in serum creatinine or creatinine CLr. However, the changes in serum creatinine associated with OCT2/MATEs DDIs are frequently insufficient to support its use as a biomarker [134]. 

In recent studies, m^1^A has been incorporated as a novel biomarker for OCT2 and MATE1/2-K. To extend the understanding of endogenous probes for OCT2/MATEs, Miyake, T. et al. [135] first investigated novel endogenous substrates of OCT2 by the metabolomic analysis of plasma and urine samples from wild-type and OCT1/2 double-knockout mice. After verifying the transport of the candidate compound m^1^A by OCT2/MATEs, the researchers evaluated its utility as an alternative probe for clinical DDI studies in animals and humans. The results showed that m^1^A could be an alternative probe for evaluating DDIs involving OCT2 and MATE1/2-K. In addition, Miyake, T. et al. [124] also demonstrated m^1^A as a superior OCT2 and MATE1/2-K biomarker through a series of experiments in another study.

The use of endogenous biomarkers has been expanded to complex DDIs with several possible interaction sites. From these, DDI risks involving multiple drug transporters can be evaluated in the same projects. The plasma or urinary levels of these biomarkers can enable the monitoring of drug transporter activities without the need to conduct a clinical DDI study. While biomarkers are effective, for some drugs, the production of biomarkers may not be obvious enough. Additionally, because of the overlap in substrate specificity for individual transporters, it is unlikely that specific biomarkers will be identified for each transporter of interest [136,137].

### 5.6. In Silico Models

In silico models have been used to predict DDIs, as it is impossible to test all possible drug combinations through experiments. Currently, the in silico models for predicting DDIs mediated by renal transporters mainly include machine learning methods (MLMs) and physiologically based pharmacokinetic (PBPK) models. The following is a discussion of these two models.

#### 5.6.1. MLMs

MLMs are a type of ligand-based computational method used for classification interactions with target proteins. Examples of MLMs include random forest (RF), support vector machine (SVM), recursive partitioning (RP), k-nearest neighbor (K-NN), and Bayesian models [138,139]. Recently, MLMs represented by Bayesian models have been used to predict renal transporter-mediated DDIs. Sandoval, P. J. et al. [140] examined the inhibitory efficacy of 400 or more compounds against the OCT2-mediated uptake of six structurally different substrates. Discovery Studio version 4.1 (Biovia, San Diego, CA, USA) was used to generate and validate Laplacian-corrected naive Bayesian classifier models. The same threshold was used (50% inhibition or higher), as well as the same method of 5-fold cross-validation and receiver operating characteristic (ROC) calculation. Testing data sets containing 80 compounds were collated to assess the predictive capacity of training data and generate statistical results. These datasets have been used for the development of Bayesian machine learning analysis and prediction algorithms. In early preclinical drug discovery, it is possible to use the virtual screening of large libraries of novel structures to identify potential OCT2 interactions. This approach can be a cost-effective way to eliminate compounds that may pose a problem. The above-mentioned study demonstrated how substrate-specific OCT2 data were used to generate predictive computational models. In another study by the same experimental group, Martinez-Guerrero, L. et al. [141] used a corresponding array of Bayesian models to generate predictions for each compound. 

In summary, Bayesian models have the advantage of being fast and interpretable by defining toxicological vectors in fingerprint features. In early drug discovery, MLMs can quickly predict the substrate or inhibitory properties of candidate drugs. These methods have the potential to help identify DDI perpetrators among existing drugs and guide further experimental validation [116,142]. MLMs have the following advantages: handling complex data, automated analysis, and strong predictive ability. However, the need for large amounts of data is a disadvantage of MLMs.

#### 5.6.2. PBPK Models

PBPK models are mathematical models based on anatomy, biochemistry, and physicochemical principles. PBPK models connect a series of physiologically meaningful compartments or organs through the circulation system, describing the ADME process of drugs in the body. Currently, PBPK models have had important impacts on drug development and post-marketing stages. The latest FDA guidance on drug interactions recommends the use of PBPK modeling approaches to predict potential drug interactions. PBPK models can predict DDIs caused by different conditions, different individuals, and different mechanisms, and thus guide the development of further experiments or substitute some DDI clinical trials [143,144,145]. PBPK models can be constructed using mathematical computing software, and most investigations into transporter DDIs have been carried out using PKPB software, such as Symcyp (Certara, Sheffield, UK), GastroPlus (simulation plus, PA, Lancaster, CA, USA), MATLAB (MathWorks, Natick, MA, USA), Pk-Simand MoBi (Bayer Technology Services, Leverkusen, Germany) [146]. The following disadvantages need to be overcome when using PBPK models: ① the complexity of parameter acquisition and calculation and ② the interpretation of the meaning of the results being relatively complex.

To quantitatively forecast DDI between cimetidine and metformin using in vitro inhibition constant (Ki) values, researchers created a new PBPK model of metformin. With in vitro Ki values, this model successfully reproduced the DDI between cimetidine and metformin. They concluded that the interaction between cimetidine and metformin is probably caused by the inhibition of MATEs by cimetidine, and not by OCT2 inhibition [147]. Asaumi, R. et al. [148] developed a rifampicin PBPK model to forecast P-gp-mediated DDIs. This PBPK model accurately predicted P-gp-mediated DDIs with talinolol, digoxin, and quinidine in a variety of situations with varying dosages, durations, timings, and delivery methods. These results show that their rifampicin PBPK model may be used to predict DDIs with different P-gp substrates and investigate the number of DDIs in the intestine, liver, and kidneys.

## 6. Future Prospects

Despite the identification of many transporters in the human kidney, a complete evaluation of the therapeutic drugs’ substrate and inhibitor affinities against these renal transporters has not been established yet. Furthermore, the in vivo significance of many kidney transporters remains uncertain. A thorough understanding of the precise transport mechanism and membrane localization within the nephron is critical to achieving a better understanding of the in vivo function of kidney transporters [149]. In addition, the kidney is also an important extrahepatic metabolic tissue in the human body, containing various phase I and II metabolic enzymes. Therefore, we need to consider not only the individual actions of renal transporters and metabolic enzymes, but also their coordinated effects. If DDIs involve both metabolic enzymes and transporters, it will bring challenges to the study of DDIs. Therefore, in the future, better evaluation methods need to be found to comprehensively evaluate DDIs.

With the emergence of new drug development technologies and advancements in molecular biology and computer technology, there will be more efficient and sensitive methods for studying transporters. These new methods will provide a more comprehensive and scientific basis for understanding the in vivo processes of new compounds, structural modifications, transporter-mediated drug interactions, improving drug bioavailability, reducing adverse drug reactions, and achieving rational clinical use of drugs.

## 7. Conclusions

Quantitative prediction of renal transporter-mediated DDIs is essential for the prevention of adverse drug reactions and the rationalization of clinical study programs. In this paper, the commonly used methods for the study of renal transporter-mediated DDIs are sorted and presented in a timely manner with the latest research progress, and their respective advantages and disadvantages are summarized. From this paper, researchers at different stages of study can choose a suitable research method for their own study of certain renal transporter DDIs, taking into account their own subject matter and the actual situation in their laboratories. Due to the complexity of the in vivo transport mechanism and the many influencing factors, it is not enough to use a single method to understand the function of one or some transporters. It is often necessary to weigh the advantages and disadvantages of various methods, and the results obtained by combining multiple methods are corroborated with each other for comprehensive analysis and validation.

## Figures and Tables

**Figure 1 molecules-28-05252-f001:**
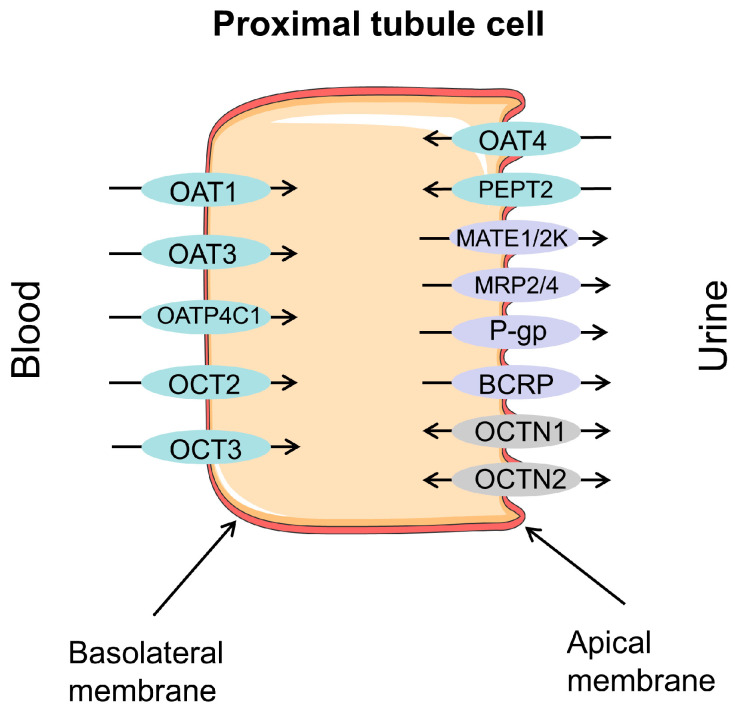
Major drug transporters in proximal tubular cells of the kidney: OAT1—organic anion transporter 1; OAT3—organic anion transporter 3; OAT4—organic anion transporter 4; OATP4C1—organic anion transporter polypeptide 4C1; OCT2—organic cation transport 2; OCT3—organic cation transport 3; PEPT2—peptide transporter 2; MATE1—multidrug and toxin extrusion protein 1; MATE2-K—multidrug and toxin extrusion protein 2 kidney-specific; MRP2—multidrug resistance-associated protein 2; MRP4—multidrug resistance-associated protein 4; P-gp—P-glycoprotein; BCRP—breast cancer-resistance protein; OCTN1—organic cation/carnitine transport 1; OCTN2—organic cation/carnitine transport 2. Efflux transporters/carriers highlighted in purple, influx carriers in blue, and bidirectional carriers in gray.

**Figure 2 molecules-28-05252-f002:**
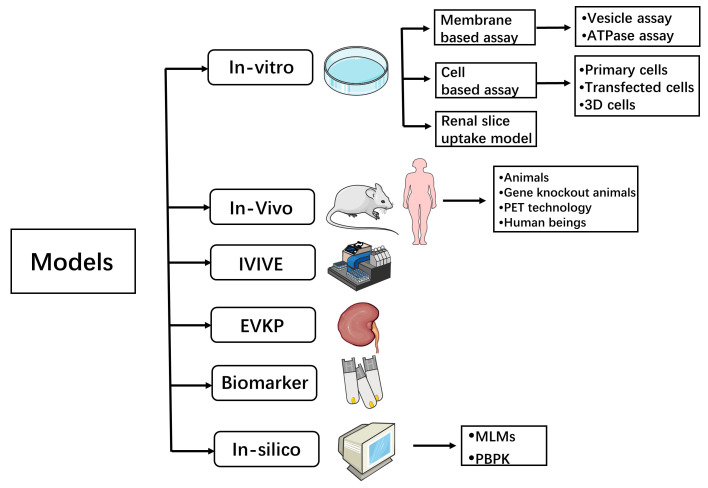
Models for transporter-mediated DDI studies. IVIVE—In vitro–in vivo extrapolation; EVKP—Ex vivo kidney perfusion; PET technology—Positron emission tomography technology; MLMs—machine learning methods; PBPK—physiologically based pharmacokinetic.

**Table 2 molecules-28-05252-t002:** Examples of classical DDIs mediated by renal transporters.

Transporter Name	Victim Drug	Perpetrator Drug	CLr Decrease (%)	References
OAT1, OAT3	Acyclovir	Benzylpenicillin	56	[68]
	Acyclovir	Probenecid	32	[65]
	Furosemide	Probenecid	>50	[69]
	Cidofovir	Probenecid	52	[70]
	Fexofenadine	Probenecid	73	[71]
OCT2, MATE1, MATE2-K	Metformin	Cimetidine	27	[72]
	Metformin	Pyrimethamine	23–35	[62]
	Pindolol	Cimetidine	34	[73]
	Procainamide	Ofloxacin	30	[74]
	Zidovudine	Trimethoprim	48	[75]
P-gp	Digoxin	Quinidine	33	[76]
	Digoxin	Verapamil	21	[67]
	Digoxin	Ritonavir	35	[77]
	Digoxin	Itraconazole	20	[78]

## Data Availability

Not applicable.

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
