# Peer review of "Research Methods and New Advances in Drug–Drug Interactions Mediated by Renal Transporters"

_molecules, 2023, doi:10.3390/molecules28135252_

Round 1

Reviewer 1 Report

The conclusions are not precise, Table 1 should appear in the results or discussion, or body of the review. abstrac poor information.

Figures or table would be missing in part 3 and 4 future prospects put it in a different place than the conclusions.  

Author Response

Dear Reviewer:

Thank you very much for your consideration and valuable suggestions to our manuscript ID molecules-2454729 entitled “Research methods and new advances in drug-drug interactions mediated by renal transporters”.

On your request, we made some further revisions. You may find our revision and incorporation of reviewers’ suggestion. The revised portions were marked in red bold.

We hope the manuscript has been improved satisfactorily and it will be accepted for publication in Molecules. We are looking forward to your positive decision.

Best regards,

Yours sincerely,

Yanna Zhu, Ph.D.

Department of Pharmacy, First Affiliated Hospital of Dalian Medical University, Dalian, China;

-Reviewer: 1

1.The conclusions are not precise, Table 1 should appear in the results or discussion, or body of the review. abstrac poor information.

Answer: Yes, thank you very much for the instruction. Special thanks to you for your helpful suggestions. Considering the suggestion, we rewrote the abstract and conclusion and marked them in red bold on page 1, line 12-18, page 16, line 714-727. Regarding the Table 1, it has been deleted considering the suggestions of reviewer 2. And a brief description of the advantages and disadvantages of each research method is provided in the body of Part 5, page 8-15 in red bold.

  1. Figures or table would be missing in part 3 and 4 future prospects put it in a different place than the conclusions.  

Answer: Yes, thank you very much for the instruction. Considering the suggestion, we added two tables in part 3 and 4, marked them in red bold on page 5, line 231; page 6, line 260. And we separate the future prospects from the conclusion, with the future prospects in Part 6, page 15-16 and the conclusion in Part 7, page 16.

Reviewer 2 Report

Review comments on molecules-2454729

Journal : Molecules (ISSN 1420-3049)

Manuscript ID : molecules-2454729

Type : Review

Title : Research methods and new advances in drug-drug interactions mediated by renal transporters

Authors : Kexin Lin , Xiaorui Kong , Xufeng Tao , Xiaohan Zhai , Linlin Lv , Deshi Dong , Shilei Yang * , Yanna Zhu *

Section : Medicinal Chemistry

Special Issue : New Advances in Drug Metabolism and Pharmacokinetics

Major comments 

Kidney is critical in the human body’s excretion of drugs and their metabolites. Renal transporters participate in actively secreting substances from the proximal tubular cells and reabsorbing them in the distal renal tubules. However, when two drugs are used simultaneously, their functions would be modulated by side effects of the other drug and may produce drug-drug interactions (DDIs) of clinical importance. Therefore, this review paper seems very important for the researchers in this field since authors describe the traditional research methods of renal transporter-mediated DDIs, including the recent progress of the methods. Accordingly, this review paper can be acceptable if following points are clarified. 

(1)   Major defect of this review paper is lacking of considerations for the structural information on the transporter molecules. Since the main theme of this review paper is drug-drug interactions (DDIs), such drugs may compete on the same site or may modulate each other for the transport of drugs by binding to the allosteric site of the same transporter molecule. Accordingly, structural information is highly necessary. As described by the authors, most of drug transporters are a member of either SLC superfamily or ABC superfamily and therefore, structural information of these protein molecules in other species would be available.

(2)   Table 1 is not necessary. This summarized table gave similar but vague and redundant information for each experimental model, which is described in the main text concisely. Authors should add more detailed description for the cited studies in the main text, which would be more beneficial for the potential readers. 

Minor comments

(1)     “can provide energy by hydrolyzing ATP to transport drugs out of cells” (page 2, line 61) is misleading. It might be better to change “can utilize energy by hydrolyzing ATP to transport drugs out of cells”

(2)      “et al” (page 5, line 204) might be “etc”?

(3)   Figure 1 needs some additional captions otherwise it is difficult to understand.

(4)   For Figure 2, some additional captions may be added. 

(5)   “K, P.A. et al [94]” (page 10, line 439) is incorrect. It should be Arun et al [94].

This is due to the mistakes in ref #94 (page 21, line 875). Some Indian researchers use their own names as family name first and, then, given name. Therefore, the reference for [94] should be corrected for other coauthors. For other references in the list, it seems OK.

Author Response

Dear Reviewer:

Thank you very much for your consideration and valuable suggestions to our manuscript ID molecules-2454729 entitled “Research methods and new advances in drug-drug interactions mediated by renal transporters”.

On your request, we made some further revisions. You may find our revision and incorporation of reviewers’ suggestion. The revised portions were marked in red bold.

We hope the manuscript has been improved satisfactorily and it will be accepted for publication in Molecules. We are looking forward to your positive decision.

Best regards,

Yours sincerely,

Yanna Zhu, Ph.D.

Department of Pharmacy, First Affiliated Hospital of Dalian Medical University, Dalian, China;

-Reviewer: 2

Major comments: 

Kidney is critical in the human body’s excretion of drugs and their metabolites. Renal transporters participate in actively secreting substances from the proximal tubular cells and reabsorbing them in the distal renal tubules. However, when two drugs are used simultaneously, their functions would be modulated by side effects of the other drug and may produce drug-drug interactions (DDIs) of clinical importance. Therefore, this review paper seems very important for the researchers in this field since authors describe the traditional research methods of renal transporter-mediated DDIs, including the recent progress of the methods. Accordingly, this review paper can be acceptable if following points are clarified.

  1. Major defect of this review paper is lacking of considerations for the structural information on the transporter molecules. Since the main theme of this review paper is drug-drug interactions (DDIs), such drugs may compete on the same site or may modulate each other for the transport of drugs by binding to the allosteric site of the same transporter molecule. Accordingly, structural information is highly necessary. As described by the authors, most of drug transporters are a member of either SLC superfamily or ABC superfamily and therefore, structural information of these protein molecules in other species would be available.

Answer: Yes, thank you very much for the instruction. Considering the suggestion, we added structural information about the transporter in Part 2, page 3-5 and marked it in red bold.

  1. Table 1 is not necessary. This summarized table gave similar but vague and redundant information for each experimental model, which is described in the main text concisely. Authors should add more detailed description for the cited studies in the main text, which would be more beneficial for the potential readers. 

Answer: Yes, thank you very much for the instruction. Considering the suggestion, we deleted Table 1, and a brief description of the advantages and disadvantages of each research method is provided in the body of Part 5, page 8-15 in red bold.

Minor comments:

1.“can provide energy by hydrolyzing ATP to transport drugs out of cells” (page 2, line 61) is misleading. It might be better to change “can utilize energy by hydrolyzing ATP to transport drugs out of cells”.

Answer: Thank you very much for your reminder. We feel sorry for our carelessness. We corrected the provide ” into “utilize” and marked it in red bold on page 2, line 67-68.

  1. “et al” (page 5, line 204) might be “etc”?

Answer: Thank you very much for your reminder. We feel sorry for our carelessness. We corrected the et al ” into “ etc ” and marked it in red bold on page 6, line 257.

  1. Figure 1 needs some additional captions otherwise it is difficult to understand.

Answer: Yes, thank you very much for the instruction. Considering the suggestion, we added some additional captions to Figure 1 and marked them in red bold on page 3, line 79-87.

  1. For Figure 2, some additional captions may be added.

Answer: Yes, thank you very much for the instruction. Considering the suggestion, we added some additional captions to Figure 2 and marked them in red bold on page 7-8, line 275-277.

  1. “K, P.A. et al [94]” (page 10, line 439) is incorrect. It should be Arun et al [94].

This is due to the mistakes in ref #94 (page 21, line 875). Some Indian researchers use their own names as family name first and, then, given name. Therefore, the reference for [94] should be corrected for other coauthors. For other references in the list, it seems OK.

Answer: Yes, thank you very much for the instruction. Considering the suggestion, we corrected the K, P.A. ” into “ Arun ” and marked it in red bold on page 12, line 521.

We corrected other coauthors in the reference and marked it in red bold on page 24, line 1040-1041.

In molecules-2454729-review PDF, every detail mentioned by the reviewer that needs to be modified, we have modified it item by item and marked them in red bold. Based on the second reviewer's suggestion of "Minor editing of English language required", we carefully checked the English language and asked a colleague who is fluent in English writing to check my manuscript. Thanks again for the reviewer's affirmation and modification of the manuscript.

Round 2

Reviewer 1 Report

Thank you for the changes made, to improve the work presented.
The changes made have improved the work.  

Reviewer 2 Report

Review comments on molecules-2454729-revised version

Journal : Molecules (ISSN 1420-3049)

Manuscript ID : molecules-2454729

Type : Review

Title : Research methods and new advances in drug-drug interactions mediated by renal transporters

Authors : Kexin Lin , Xiaorui Kong , Xufeng Tao , Xiaohan Zhai , Linlin Lv , Deshi Dong , Shilei Yang * , Yanna Zhu *

Section : Medicinal Chemistry

Special Issue : New Advances in Drug Metabolism and Pharmacokinetics

Major comments 

I have checked the revised version of molecules-2454729 thoroughly.  In the revised version, authors made extensive and appropriate revisions and changes against the comments and suggestions raised by reviewer.  Importantly, authors removed table (old Table 1) and added new tables (Table 1 and Table 2). New Table 1 summarizes transporter expression changes across kidney diseases. New Table 2 summarizes classical DDIs mediated by renal transporters. These new tables will help the understandings of potential readers in this field. The responses indicated in the cover letter are also very reasonable. Therefore, I believe that this revised manuscript has been sufficiently improved to warrant publication in “molecules” in the present form. 
